# Atomic-level handedness determination of chiral crystals using aberration-corrected scanning transmission electron microscopy

Zhuoya Dong[1] & Yanhang Ma [1✉]

Handedness or chirality determination is a challenging and important topic in various fields including chemistry and biology, as two enantiomers have the same composition and mirror symmetry related structures, but might show totally different activities and properties in enantioselective separations, catalysis and so on. However, current methods are unable to reveal the handedness locally of a nanocrystal at the atomic-level in real-space imaging due to the well-known fact that chiral information is lost in a two-dimensional projection. Herein, we present a method for handedness determination of chiral crystals by atomic-resolution imaging using Cs-corrected scanning transmission electron microscopy. In particular, we demonstrate that enantiomorphic structures can be distinguished through chirality-dependent features in two-dimensional projections by comparing a tilt-series of high-resolution images along different zone axes. The method has been successfully applied to certify the specific enantiomorphic forms of tellurium, tantalum silicide and quartz crystals, and it has the potential to open up new possibilities for rational synthesis and characterization of chiral crystals.

[1] School of Physical Science and Technology, ShanghaiTech University, 201210 Shanghai, China. ✉email: mayh2@shanghaitech.edu.cn

Chirality is a common phenomenon in nature. Many drugs and inorganic materials crystallize into two forms that are related by mirror symmetry. Particular properties, such as optical activity, piezoelectricity, chiral dichroism, enantioselective separations, and catalysis, happen only in chiral crystals. The importance of chirality can be seen in the pharmaceutical industry, where two enantiomers of chiral drugs might show totally different activities. One example is the well-known drug disaster caused by side-effects of thalidomide, for which, one enantiomer is a safe sleeping pill while the other can lead to deformity. Therefore, determination of chirality is one of the most important issues in crystallography. Single-crystal X-ray diffraction has been the most widely used technique to distinguish between enantiomorphic forms by comparing reflection intensities of Bijvoet pairs rising from anomalous scattering[1,2]. However, determination of chirality by X-ray diffraction usually needs a large-sized single crystal of high quality that is free from defects.

Electron microscopy (EM) is a powerful tool to provide structural information, even for nanocrystals. Compared with X-rays, electron beams have a much stronger interaction with matter, and the theoretical resolution achieved by using high-energy electrons is also higher than that of laboratory X-ray's. In electron diffraction, chirality determination is mainly based on the intensity asymmetry of Bijvoet pairs caused by multiple-beam scattering[3]. The most developed electron crystallography method is the convergent beam electron diffraction (CBED) technique[3–10]. The precession electron diffraction (PED) technique, which has the same principle as CBED for determining chirality, has also been successfully applied to electron-beam-sensitive crystals, such as zeolites[11]. Recently, the absolute configuration of a chiral pharmaceutical organic crystal was determined using precession electron diffraction tomography (PEDT) combined with dynamical refinement[12]. However, electron diffraction provides averaged structural information from many unit cells, so it is not applicable to crystals with poor crystallinity or defects. Besides, electron diffraction are usually sensitive to variations in crystal thickness, mainly due to multiple scattering effects, and thus it demands very careful data collection and processing. High-resolution transmission electron microscopy (HRTEM) allows direct observation of a two-dimensional (2D) projected crystal structure at the atomic-level under certain conditions, such as a thin crystal and close to Scherzer defocus. By comparing the difference of two HRTEM images taken from different zone axes, the handedness of small zeolite crystals has been determined[11]. The limitation of HRTEM is that the contrast observed in images is strongly affected by the contrast transfer function (CTF), so it may not give the correct representation of the crystal structure under some conditions. Other methods, such as electron backscatter diffraction[13], fringes in TEM images[14], and electron vortex beams[15] have been used for the determination of handedness. These methods, however, are either effective only for special cases or require complicated procedures, and none of them can be applied for local handedness determination at the atomic-level. Therefore, developing new electron crystallographic methodology for determining the handedness of crystals locally is a challenging but important task.

Scanning transmission electron microscopy (STEM) has become a widely used technique for atomic-scale analysis in recent years. With aberration corrections, the resolution of STEM has been remarkably improved, reaching the sub-angstrom scale[16]. More importantly, STEM images do not suffer from the conventional CTF effect, thus showing contrast robustness against the variation of focus and specimen thickness, and are therefore easier to be interpreted compared to conventional TEM images[17]. Thus, STEM has the potential to determine crystal handedness at the atomic-level, which is beneficial for crystals with poor crystallinity or defects, especially when the two arrangements with opposite handedness occur together in one single crystal.

Here, we report an electron crystallography method using aberration-corrected STEM to determine handedness locally at the atomic-level. A set of two STEM images from the same crystal are taken by tilting the crystal clockwise or anticlockwise from one axis to another around a certain direction. The comparison of two enantiomorphic structures along certain zone axes reveals a clear difference, which shows mirror symmetry related atomic arrangements. Handedness can be directly determined by observing the specific atomic arrangements in STEM images. Furthermore, tilt series of STEM images along different axes were simulated to corroborate our new method. This method has been successfully applied to the handedness determination of chiral tellurium, tantalum silicide and quartz crystals.

## Results

**Crystal structure of tellurium.** Tellurium (Te) is a typical elemental semiconductor with a narrow band gap of 0.35 eV at room temperature[18]. The relatively heavy atomic mass and intrinsically nested valence band structure of tellurium make it a high-performance elemental thermoelectric material[19]. It has also been studied in regard to other physical properties, such as photoconductivity, piezoelectricity, and catalytic activity[20–23]. The structure of trigonal tellurium consists of parallel helical Te chains packed in a hexagonal array running along the $c$ axis (Fig. 1). The configuration of helical chains leads to opposite handedness with different space groups, the right-handed one corresponds to space group $P3_121$ (No.152) and the other to $P3_221$ (No.154). The structure also has twofold rotational symmetry along the $a$, $b$, and $a + b$ directions. Unit cell parameters at 300 K are $a = b = 4.458$ Å, $c = 5.925$ Å, $\alpha = \beta = 90°$, $\gamma = 120°$. In right-handed Te (space group $P3_121$), the atom position is at the site $3a$ $(x, 0, \frac{1}{3})$, where $x = 0.745$[24]. Each atom forms strong covalent bonds with its two nearest neighbors in the helical chains, while adjacent chains are bound together by weak van der Waals interactions, which makes Te inherent chiral[25]. Chiral tellurium crystals are predicted to possess $p$-type thermoelectric transport properties and anisotropic lattice thermal conductivity, both of which have a strong relationship with the anisotropy parallel and perpendicular to the helical chains[25,26]. Other properties, such as the transition to a strong topological insulator under strain, strong electrical magnetochiral anisotropy and band splitting, are all related to the chirality of tellurium[27–29].

**Handedness determination by STEM imaging.** First, we explore whether the two enantiomorphic structures of Te can be distinguished from structural models, which provides the theoretical support for this crystallography method. The two chiral Te

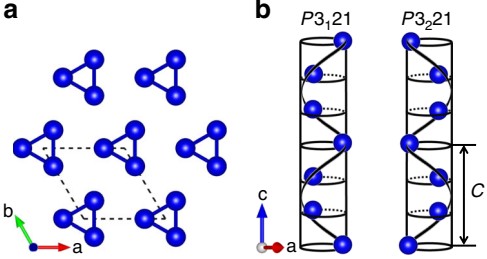

**Fig. 1 Crystal structure of Te. a** The structure consists of helical Te chains packed in a hexagonal array running along [001] direction. **b** Helical chains of Te atoms run in different manners for two enantiomorphic structures.

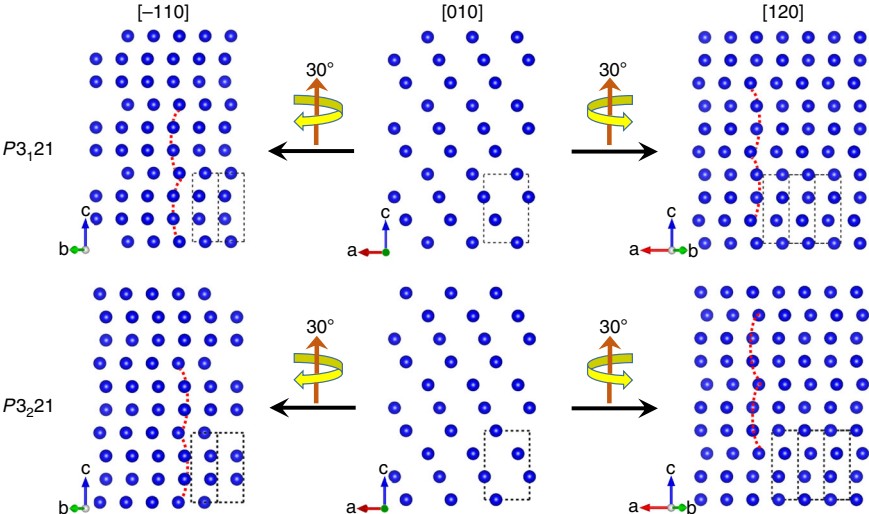

**Fig. 2 A tilt-series of Te structural models with $P3_121$ and $P3_221$ space groups.** Projections of Te crystal structural models with right-/left-handedness along different orientations in a tilt series.

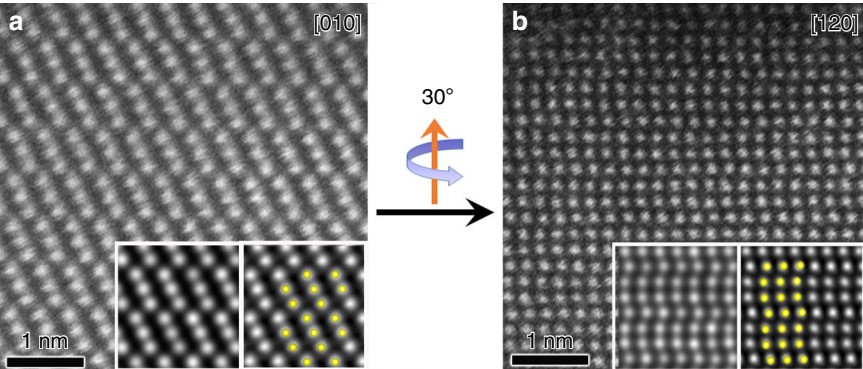

**Fig. 3 Handedness determination of one Te crystal.** STEM-ADF images of a Te crystal along **a** [010] and **b** [120] zone axes in a tilt-series (insets are simulated (left) and $p1$ symmetry-averaged images (right) overlaid with structural models, with yellow spheres representing Te atoms).

structures were oriented along the [010] direction, and apparently, it is not possible to distinguish them from the projections of the two models (Fig. 2). Then, the structure was oriented along the [−110] direction by rotating the models from [010] direction clockwise (with the $c$ axis up) by 30° around the spiral axis. The curves of the atomic arrangement along the $c$ axis bend towards opposite directions for the two enantiomorphic structures, where the atomic chains bend to the left and right for the right-handed and left-handed structures, respectively. Similarly, the structure reaches the [120] zone axis by rotating the models from the [010] direction anticlockwise by 30° around the spiral axis. In this case, curves in the right-handed structure bend to the right and curves in the left-handed structure bend to the left. Thus, recording two images along proper zone axes in a tilt series is sufficient to determine the handedness of one chiral crystal by observing the chirality-dependent features of the atomic arrangements along certain directions. Of note, when a right-handed tellurium crystal is rotated by 30° clockwise with the rotation axis up, the bending direction is always to the left, irrespective of the $c$ axis direction (see Supplementary Fig. 1).

In order to confirm the experimental feasibility of this method, a single tellurium crystal was studied using a probe-corrected STEM. The crystal was first aligned along the [010] zone axis by tilting the $\alpha$ and $\beta$ axes of the goniometer under selected area electron diffraction (SAED) mode. Then, the microscope was switched to STEM mode and STEM-ADF images were recorded

with resolution better than 0.9 Å. After that, the same crystal was continuously tilted anticlockwise by 30° around the $c$ axis. When the [120] zone axis was reached, the corresponding STEM-ADF image was taken (Fig. 3). High-resolution images were obtained in both cases, which can resolve a single Te atomic column. The Z-contrast images can be directly interpreted, with bright spots in ADF representing Te atoms. The atomic arrangement running along the $c$ axis in Fig. 3b is readily recognized to be bended to the left, which implied a left-handed Te crystal structure.

The selection of zone axes in a tilt series will vary depending on the crystal structure. For the trigonal crystal system, it is possible to use [−120] and [130] zone axes by tilting a crystal from [010]. Atomic chains along the $c$ axis in these two projections also show opposite bending directions for two enantiomorphic structures (see Supplementary Fig. 2). Besides, in the trigonal system, the [100] and [010] axes are symmetrically equivalent. Thus, a tilt series starting from [100] or [010] gives the same result (see Supplementary Figs. 3 and 4).

In previous work, we have successfully used HRTEM imaging to determine the handedness of a chiral zeolite nanocrystal by observing the shift direction in a set of two HRTEM images through rotating the crystal around a screw axis[11]. There are actually two preconditions for that method: first, obvious features such as layers with strong contrast should be selected as references to distinguish the shift direction; second, the origin (height) of the crystal should match in both images, therefore

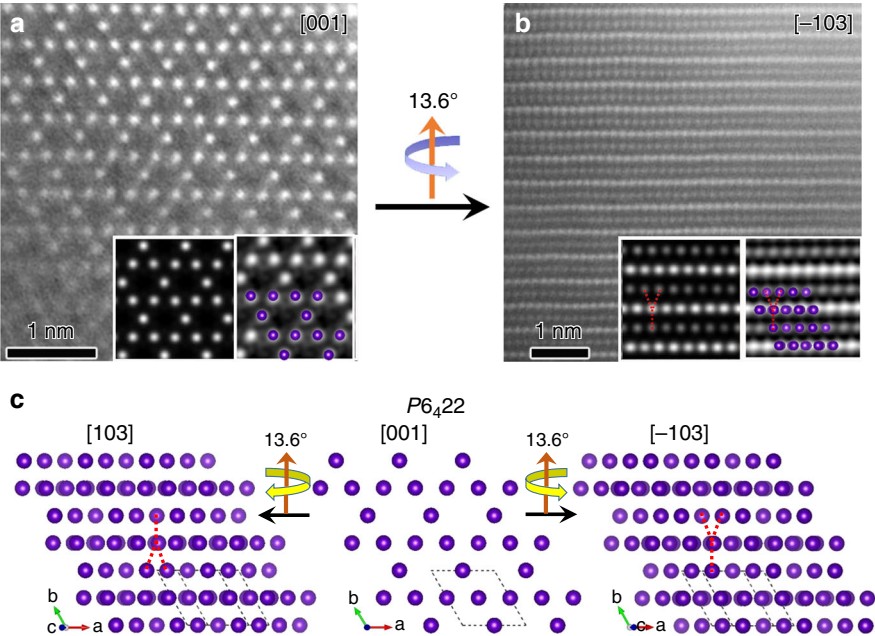

**Fig. 4 Handedness determination of one TaSi2 crystal.** STEM-ADF images of a tantalum silicide crystal along the **a** [001] and **b** [−103] zone axes (insets are simulated (left) and $p$1 symmetry-averaged images (right) overlaid with structural models; purple spheres represent Ta atoms, Si atoms are omitted for clarity). **c** Projections of TaSi$_2$ crystal structural models with left-handedness along different orientations in a tilt series.

gold nanoparticles were used to align two images taken along different zone axes. However, the alignment using gold nanoparticles will fail at atomic-level resolution, and the resolution of HRTEM images of the chiral zeolite was also limited (around 1.6 Å) due to the electron bean damage at high magnification. Herein, the new approach determines a specific handedness using chirality-dependent features in two-dimensional projections by comparing a tilt-series of high-resolution images taken along different zone axes. As a result, neither references nor origin alignment are needed. Besides, STEM with aberration-correction offers much higher resolution that can reach sub-angstrom, and it usually does not suffer contrast reversing caused by crystal thickness variations or focus changing of lens (see Supplementary Fig. 5). HRTEM and STEM images of Te crystal are compared in Supplementary Fig. 6.

**Method application.** To further extend the scope of this method, tantalum silicide (TaSi$_2$), a binary transition-metal silicide with the hexagonal C40 structure, was studied. The C40 TaSi$_2$ structure consists of hexagonally arranged TaSi$_2$ layers stacked along the $c$ axis. In each layer, one Ta atom is coordinated with six Si atoms, while one Si atom is coordinated with three Ta atoms and three Si atoms. The stacking order of the layers can be ABC and CBA, which correspond to space groups $P6_2$22 (No.180) and $P6_4$22 (No.181), respectively[30]. The unit cell parameters for left-handed ($P6_4$22) TaSi$_2$ are $a = b = 4.78$ Å, $c = 6.57$ Å, $\alpha = \beta = 90°$, $\gamma = 120°$, respectively[31]. In this case, the projections along [120] and [−110] show no difference as the structure has a six-fold screw axis along the $c$ direction (see Supplementary Fig. 7). Instead, a set of STEM-ADF images from a single crystal were recorded by tilting the crystal from the [001] direction to [−103] direction around [120] axis (same direction as $b^*$ axis in reciprocal space) by about 13.6° (Fig. 4). A single Ta atomic column can be easily resolved, while different contrast of the Ta columns arranged alternately along the [120] axis is due to the different numbers of Ta atoms in adjacent layers in the projection. Each brighter Ta column forms a "Y" shape with its nearby two upper atomic column and one lower column, which is the

feature of the left-handed structure. The experimental results are also consistent with simulated STEM images in a tilt series (see Supplementary Fig. 8). Of note, the tilting of TaSi$_2$ from [001] to [−103] symmetrically equivalent directions always leads to a "Y" shape feature for the left-handed structure and an inverted "Y" shape feature for the right-handed structure (see Supplementary Fig. 9). Therefore, the structure of TaSi$_2$ in Fig. 4 is left-handed with space group of $P6_4$22.

Quartz is a natural mineral that is a prototypical example of chiral materials. The quartz structure also belongs to the trigonal crystal system with chiral space groups $P3_1$21 (No.152) or $P3_2$21 (No.154). We have also studied quartz using an experimental process similar with that for tellurium. Zone axes were reached under TEM mode using SAED (see Supplementary Fig. 10). A set of STEM-ABF images from one single crystal were recorded by clockwise tilting of the crystal by 30° from the [110] direction to the [120] direction. In this case, instead of ADF images, STEM-ABF images were used to give better contrast of light elements. However, compared with tellurium and tantalum silicide, quartz consists of lighter elements (silicon and oxygen) and is more sensitive to electron beams. As a result, the resolution of tilt series STEM images is limited, because of serious beam damage. The handedness can be determined with the help of symmetry-averaged images, but not at atomic resolution (see Supplementary Figs. 11–14).

In order to prove the universality of this method, tellurium oxide (TeO$_2$, space group $P4_1$2$_1$2/$P4_3$2$_1$2) and $\beta$-glutamic acid (space group $P2_1$2$_1$2$_1$) were also studied based on structural models and simulated images (see Supplementary Figs. 15 and 16). Differences can be always found in a tilt series of images to distinguish the two enantiomorphic structures.

**Discussion**

In summary, a new method has been developed to determine the local handedness of chiral crystals at the atomic-level by using aberration-corrected STEM. The two enantiomorphic structures can be distinguished through chirality-dependent features in two-dimensional projections by comparing a tilt-series of high-

resolution images along different zone axes. The method has been successfully applied to certify specific enantiomorphic forms of tellurium, tantalum silicide, and quartz crystals through experiments. The generality of this method has also been demonstrated in other cases through structure modelling and simulations. This electron crystallography method provides a new way for handedness determination locally of nanosized crystals at atomic resolution. It also shows potential applications in the study of racemates containing a mixture of two domains with different handedness.

## Methods

**Materials**. All reagents were commercially available and used without further purification. Tellurium, tantalum silicide, and quartz samples were all purchased from Aladdin. Bulk crystals were crushed, dispersed in ethanol, sonicated, and few drops of the suspension were placed onto holey carbon copper grids.

**Experiments and characterization**. STEM measurements were performed with a cold-FEG aberration-corrected JEOL Grand ARM 300 (JEOL Ltd.) operating at 300 kV acceleration voltage. The instrument is equipped with double Cs-correctors. The microscope in STEM mode have two annular dark field (ADF) detectors and an annular bright field (ABF) detector. The crystallinity and phase purity of the powder samples were characterized by powder X-ray diffraction (PXRD) on a Bruker D8 Advance X-ray diffractometer (40 kV, 40 mA) using Cu Kα radiation ($\lambda = 1.5418$ Å). PXRD patterns are shown in Supplementary Fig. 17.

**STEM simulation**. STEM image simulations were performed using a free software package QSTEM (http://www.qstem.org), which is based on the multi-slice algorithm. Simulation parameters were roughly the same for all the crystals. The simulation used the STEM mode and included thermal diffuse scattering by the frozen phonon approximation. A $80 \times 80$ pixel area and a probe array of $400 \times 400$ pixel area were employed. Box mode with a size of 60 Å × 60 Å × 100 Å ($x \times y \times z$) was used and the zone axes were reached by sample tilt function. In total, there were 100 slices along the zone axes, each slice was of one angstrom thick (50 slices with two angstrom thick per slice for $TaSi_2$). The defocus, astigmatism, and spherical aberration C3 were all set to zero in order to get the closest representation of the real structure models. The inner and outer angles for detector 1 and detector 2 were 42°, 180°, and 7.8°, 16°, respectively, which were consist with the experimental parameters.

**Image processing**. Raw STEM images were filtered using winner filter and ABSF filter realized by Digital Micrograph (https://www.gatan.com/). $P1$ symmetry-averaged images were obtained through Fourier analysis of raw images. Fourier coefficients were obtained by applying Fourier transform to the raw images and then used to get symmetry-averaged images through inverse Fourier transform.

## Data availability

The CIFs (Crystallography Information File) used in the whole text and supplementary information were downloaded from the FIZ Karlsruhe—Leibniz Institut for Information Infrastructure (ICSD, https://icsd.products.fiz-karlsruhe.de/) and Cambridge Crystallographic Data Centre (CCDC, free for charge at https://www.ccdc.cam.ac.uk). The deposition codes were ICSD 27745 and ICSD 16331 for quartz; ICSD 43596 and ICSD 96028 for $TaSi_2$; ICSD 202792 and ICSD 62898 for $TeO_2$; CCDC 1206528 and CCDC 1515814 for Glutamic Acid; CIF of $P3_221$ Te was created from CIF of $P3_121$ Te (ICSD 96502) by changing atomic coordinates (x, y, z) to (−x, −y, −z). All other data that support the findings of this study are available within the paper and its supplementary information files.

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

## Acknowledgements
This work is supported by Shanghai Pujiang Program (17PJ1406400), the Young Elite Scientist Sponsorship Program by CAST (2017QNRC001), National Key Research and Development Program of China (21835002), the National Natural Science Foundation of China (21875140) and CℏEM, SPST of ShanghaiTech University (#EM02161943). We thank Prof. Osamu Terasaki for discussions and encouragements, and Prof. Kenneth. D. M. Harris for English correction.

## Author contributions
Y.M. conceived and led this project. Y.M. and Z.D. performed the experiments and simulations together. Z.D. wrote the paper and Y.M. revised it.

## Competing interests
The authors declare no competing interests.
