## [Peer Review File · Nature Communications]

Reviewers' comments:

Reviewer #1 (Remarks to the Author):

The authors describe method of determination of handedness of quantum dots and other nanomaterials using Cs corrected scanning transmission electron microscope. The research is centered around the processing of tilt series for nanostructure of tellurium and quartz. By and large, the study is carried out well, but lacks novelty compare to earlier studies (<https://science.sciencemag.org/content/349/6245/290>); (<https://www.nature.com/articles/ncomms5302>) For that reason I cannot recommend its publication in the Nature Communications.

Reviewer #2 (Remarks to the Author):

This is a nice little paper demonstrating the use of scanning transmission electron microscopy (STEM) imaging for determining the local chirality of crystals. Although electron beam methods have been used previously, notably by one of the current authors who have used high-resolution transmission electron microscopy (HRTEM) to determine the chirality of a zeolite, here it is shown that the simpler to interpret STEM images can be used to determine chirality. I think it will be of wide-enough interest for publishing in Nature Communications and recommend publication.

In general the paper is nicely explained and the results convincing, and I require no substantive changes. I have a few minor comments that the authors might consider in any revision to their paper:

On line 31 of p2, the authors state that CBED is limited by serious beam damage to the specimen. Since STEM is also a convergent beam method, it is not clear to be that CBED should be any worse than STEM. In general, high-resolution imaging requires higher doses than diffraction.

On line 51 of p2 it is stated that markers are needed to align the two images for HRTEM. Why is this needed for HRTEM but not for STEM? This may relate to the comment on line 51 of p3. Are the markers needed to ensure a similar CTF for HRTEM rather than image alignment, as stated.

The contrast in Fig. 4 is surprisingly poor. Is there a reason for that? It should be noted that once symmetry averaging is used, local information is lost negating some of the advantages of imaging over diffraction.

Reviewer #3 (Remarks to the Author):

The manuscript presents a new method for handedness determination of two chiral nanocrystals using aberration-corrected scanning transmission electron microscopy. The crystal handedness was determined from the different characteristic features of two atomic-resolution STEM images taken along two different directions of the same crystal. The method was applied to two types of crystals, tellurium and quartz. I find that the method is clearly described and simple to apply. However, I have concerns about the generality because the two structures presented here are both very simple structures and have the same space groups. Based on the comments given below, I do not recommend the publication of the manuscript in the present form. If the authors can show that the method is generally applicable with examples of structures of different space group and complexity, it would be interesting for Nature Communications.

Below are my comments:

1. On page 2, line 18-21, the author wrote “This approach is also not applicable for elemental crystals such as tellurium and tin as all of the scattering power are uniformly affected by the imaginary part of the dispersion correction.” This is not true. The Bijvoet pairs will be different even for crystals containing single elements, and thus can be used to determine the handedness. In addition, I recommend not to use the term “elemental crystals” because it is widely used for other purposes.
2. Page 2 Line 50: The author previously reported the use of HRTEM images to determine the handedness of a zeolite nanocrystal (Ref. 11, Nat. Mat. 2017). Here they wrote “The limitation of HRTEM is that the contrast of images is affected by the contrast transfer function (CTF), so it may not be the correct reflection of crystal structure under some conditions. Moreover, the resolution is limited as markers are required to align two images.” While HRTEM images are affected by the CTF, the markers (e.g. Au nanoparticles) do not affect the resolution of the HRTEM images, and only the correct alignment of the images. In addition, “correct reflection of crystal structure” may be changed to “correct representation of crystal structure”. Furthermore, I expect that the method presented here can be also applied to HRTEM images, because the determination of handedness is through the comparison of simulated and experimental images. The effects of CTF can be taken into account in image simulations.
3. The coordinate of Te given in the manuscript is wrong. It turns out that the structure models of Te presented in the manuscript used the Te coordinate (0.737, 0, 1/3) instead of (0.263, 0, 1/3) as given in the manuscript (Line 104). I can only get the same figures as shown in Figures 1-2 as well as Supplementary Figures 1, 4-5 when using the coordinate Te: (0.737, 0, 1/3), see Figure X1. I got a different orientation of the trigonal and different handedness of the helical chain when I draw the figure using the Te coordinate (0.263, 0, 1/3), see Figure X2. It took me a lot of time to figure out this.
4. Line 49-57: The authors claimed that the current method does not need any markers, in contrast to their previous work on a chiral zeolite using HRTEM images (ref. 11). In the current work, they used the specific Te and quartz structure projections that show a defined bending direction. Will the method be applicable for other structures with different space group or more complex where atoms overlap, for example the zeolite previously published by the author?
5. The determination of the handedness of the quartz crystal is not very convincing. In Figure 4, the simulated images are compared with the symmetry-averaged images. However, the agreement of the simulated and experimental images is rather poor. The aspect a/b ratio of the unit cell in the projection differs a lot between the experimental and simulated images for both Fig. 4a and 4b. The angle between a and b axes is ca 95 degrees in the experimental image in Fig. 4b, which deviated a lot from 90 degrees in the simulated images. Furthermore, the orientations of the black features corresponding to two SiO₄ tetrahedra are different in the simulated and symmetry-average images in Figure 4a. In Figure 4b, the contrast in symmetry-averaged image differ significantly from the experimental image. The authors should mention how the zone axes were determined, what symmetry

has been applied and how this was done. Because the determination of the handedness is only based on one image of poor quality here, and a slight error in atoms positions may change the bending direction, the conclusion is not very convincing.

6. Supplementary Figure 8 should include simulated image along [110], which is relevant to the experimental image.

Figure X1. Crystal structure of Te using the Te coordinate (0.737, 0, 1/3) and space group $P3_12$ (152).

Figure X2. Crystal structure of Te using the Te coordinate (0.263, 0, 1/3) and space group $P3_12$ (152).

Point-by-Point Response to Reviewers' Comments:

Reviewer #1 (Remarks to the Author):

The authors describe method of determination of handedness of quantum dots and other nanomaterials using Cs corrected scanning transmission electron microscope. The research is centered around the processing of tilt series for nanostructure of tellurium and quartz. By and large, the study is carried out well, but lacks novelty compare to earlier studies

(<https://science.sciencemag.org/content/349/6245/290>); (<https://www.nature.com/articles/ncomms5302>)) For that reason I cannot recommend its publication in the Nature Communications.

Thanks for the reviewer's comment. In this study, we focused on local handedness determination of chiral crystals using atomic-level imaging. Although we use a tilt-series of images, the approach is quite different from TEM tomography. TEM tomography is a well-developed technique, which is used for reconstructing the 3D morphology of samples or distribution of nanoparticles in the matrix. In the two papers mentioned by the reviewer, one paper describes the 3D reconstruction of a nanoparticles with a help of graphene liquid cell (*Science*, 2015, 349, 6245). It is not related to the chirality. Besides, it also requires a very small crystal, sub-2 nm diameter Pt nanocrystal in the paper. The other one presents a 3D tomography result using dark-field stem imaging, revealing the asymmetric morphology of Te nanoparticles. However, the twist shape cannot be directly related to chirality of atomic structure. Instead, they used CD spectrum to confirm the chirality. Without CD spectrum data, they cannot claim the enantiomorphic determination only based on the 3D morphology reconstruction. Moreover, none of them really achieve atomic resolution (the Science paper claims the reconstructed features of Pt nanoparticles at near-atomic resolution). As far as we know, we are still the first who push the enantiomorphic structure determination to the sub-angstrom level using imaging.

Reviewer #2 (Remarks to the Author):

This is a nice little paper demonstrating the use of scanning transmission electron microscopy (STEM) imaging for determining the local chirality of crystals. Although electron beam methods have been used previously, notably by one of the current authors who have used high-resolution transmission electron microscopy (HRTEM) to determine the chirality of a zeolite, here it is shown that the simpler to interpret STEM images can be used to determine chirality. I think it will be of wide-enough interest for publishing in Nature Communications and recommend publication.

In general the paper is nicely explained and the results convincing, and I require no substantive changes. I have a few minor comments that the authors might consider in

any revision to their paper:

1. On line 31 of p2, the authors state that CBED is limited by serious beam damage to the specimen. Since STEM is also a convergent beam method, it is not clear to be that CBED should be any worse than STEM. In general, high-resolution imaging requires higher doses than diffraction.

Thanks for the reviewer's comment. We agree with the reviewer that both CBED and STEM use a convergent beam and high-resolution imaging usually requires higher dose than diffraction. The advantage of STEM over CBED is the direct imaging in the real space. Moreover, the electron beam is scanned over an area (the pixel time is between 9 and 30 μ s in our case) in STEM while the beam stays static for CBED pattern recording. In fact, the STEM imaging method proposed here is not particularly designed for electron beam sensitive materials as it is extremely challenging to collect atomic resolution images from beam sensitive materials. We mentioned that CBED brings beam damage to specimen previously because we compare it with precession electron diffraction using parallel beam. In the revised manuscript, the sentence "But it is limited by the serious electron beam irradiation damage to the specimen." has been deleted.

2. On line 51 of p2 it is stated that markers are needed to align the two images for HRTEM. Why is this needed for HRTEM but not for STEM? This may relate to the comment on line 51 of p3. Are the markers needed to ensure a similar CTF for HRTEM rather than image alignment, as stated.

Thanks for the reviewer's comment. In our previous work of studying chiral zeolites, we use a marker to align two images. That is because the projections of STW zeolite along [100] and [1-10] of two enantiomorphic structures look exactly same except for the shift direction along the c axis. Therefore, we cannot distinguish two structures by just comparing two images. As a result, markers are required. Similarly, in principle we can also use markers in STEM imaging but the size of markers should be very small to determine the handedness at atomic resolution as the shift is very small in the case of Te (Around 2 \AA shift after tilting from [100] to [010] by 120 degrees). Instead, herein we developed a brand new method. With atomic resolution STEM/HRTEM imaging, now it is possible to determine the handedness without markers. We choose STEM for atomic resolution imaging rather than HRTEM because HRTEM images are strongly affected by CTF and crystal thickness, as shown in Supplementary Figure 6. Of note, in principle our new method also works in HRTEM imaging with atomic resolution.

Supplementary Fig. 6 | HRTEM image and STEM-ADF image of Te crystal.

3. The contrast in Fig. 4 is surprisingly poor. Is there a reason for that? It should be noted that once symmetry averaging is used, local information is lost negating some of the advantages of imaging over diffraction.

Thanks for the reviewer's comment. The reason for that is the instability of quartz under electron beam. We have tried many times but failed to get data with better resolution. As the reviewer mentioned before, the high-resolution imaging at atomic resolution usually require high electron dose on the specimen. That's why we use symmetry averaging. The plane group symmetries along two zone axes can be obtained based on the space group. However, this method we proposed here is correct without doubt. Instead, we used another sample TaSi_2 with hexagonal crystal system to prove the feasibility and universality of our method (as shown in the new Fig. 4 in the manuscript). More examples such as TeO_2 have been given in the supporting information. The data of quartz has been moved to supporting information. With the development of new detectors, it might be possible to collect a tilt-series of atomic resolution images for handedness determination of e-beam sensitive crystals in the future.

Fig. 4 STEM-HAADF images of a tantalum silicide crystal along (a) [001] and (b) [-103] zone axes (insets are simulated (left) and $p1$ symmetry-averaged images (right) overlaid with structural models; purple spheres represent Ta atoms, Si atoms are

omitted for clarity). (c) Projections of TaSi₂ crystal structural models with left-handedness along different orientations in a tilt series.

Reviewer #3 (Remarks to the Author):

The manuscript presents a new method for handedness determination of two chiral nanocrystals using aberration-corrected scanning transmission electron microscopy. The crystal handedness was determined from the different characteristic features of two atomic-resolution STEM images taken along two different directions of the same crystal. The method was applied to two types of crystals, tellurium and quartz. I find that the method is clearly described and simple to apply. However, I have concerns about the generality because the two structures presented here are both very simple structures and have the same space groups. Based on the comments given below, I do not recommend the publication of the manuscript in the present form. If the authors can show that the method is generally applicable with examples of structures of different space group and complexity, it would be interesting for Nature Communications.

Below are my comments:

1. On page 2, line 18-21, the author wrote “This approach is also not applicable for elemental crystals such as tellurium and tin as all of the scattering power are uniformly affected by the imaginary part of the dispersion correction.” This is not true. The Bijvoet pairs will be different even for crystals containing single elements, and thus can be used to determine the handedness. In addition, I recommend not to use the term “elemental crystals” because it is widely used for other purposes.

Thanks for the reviewer’s comment. We are sorry for that. The sentence “This approach is also not applicable for elemental crystals such as tellurium and tin as all of the scattering power are uniformly affected by the imaginary part of the dispersion correction.” has been deleted.

2. Page 2 Line 50: The author previously reported the use of HRTEM images to determinate the handedness of a zeolite nanocrystal (Ref. 11, Nat. Mat. 2017). Here they wrote “The limitation of HRTEM is that the contrast of images is affected by the contrast transfer function (CTF), so it may not be the correct reflection of crystal structure under some conditions. Moreover, the resolution is limited as markers are required to align two images.” While HRTEM images are affected by the CTF, the markers (e.g. Au nanoparticles) do not affect the resolution of the HRTEM images, and only the correct alignment of the images. In addition, “correct reflection of crystal structure” may be changed to “correct representation of crystal structure”. Furthermore, I expect that the method presented here can be also applied to HRTEM images, because the determination of handedness is through the comparison of

simulated and experimental images. The effects of CTF can be taken into account in image simulations.

Thanks for the reviewer's comment. The new method reported here is different from that we published before. In our previous method: (1) the resolution is limited and we cannot collect atomic resolution imaging from chiral zeolite even using a Cs-corrected TEM; (2) the markers must be used because the projections along [100] and [1-10] of two enantiomorphic structures looks exactly same except for the shift direction along the c axis. The previous method works for chiral zeolite because the chiral zeolite has large lattice constant. The shift difference between two enantiomorphic structures is around 5 Å by tilting from [100] to [1-10]. Therefore, a gold nanoparticle can be used as marker to align two images. However, if we want to use a marker here to distinguish structures at sub-angstrom level. It works in principle but we might need a marker with sub nanometer size. We cannot exclude the possibility but it will be very difficult in practice.

The reviewer's expectation is correct. That is true that our new method can be also applied to HRTEM imaging by taking CTF and crystal thickness into account. However, the contrast of HRTEM imaging is much complicated compared STEM imaging when it reaches to atomic resolution, as shown in the supplementary Figure 6. The contrast changes a lot even in the thinner area of the crystal.

Supplementary Fig. 6 | HRTEM image and STEM-ADF image of Te crystal.

3. The coordinate of Te given in the manuscript is wrong. It turns out that the structure models of Te presented in the manuscript used the Te coordinate (0.737, 0, 1/3) instead of (0.263, 0, 1/3) as given in the manuscript (Line 104). I can only get the same figures as shown in Figures 1-2 as well as Supplementary Figures 1, 4-5 when using the coordinate Te: (0.737, 0, 1/3), see Figure X1. I got a different orientation of the trigonal and different handedness of the helical chain when I draw the figure using the Te coordinate (0.263, 0, 1/3), see Figure X2. It took me a lot of time to figure out this.

Thanks for the reviewer's comment. The wrong coordinates were used in the

main text. We are sorry for this error. We have corrected it in the main text.

4. Line 49-57: *The authors claimed that the current method does not need any markers, in contrast to their previous work on a chiral zeolite using HRTEM images (ref. 11). In the current work, they used the specific Te and quartz structure projections that show a defined bending direction. Will the method applicable for other structures with different space group or more complex where atoms overlap, for example the zeolite previously published by the author?*

We have conducted more experiments on other materials, as shown in the Fig. 4 and Supporting information. TaSi₂ has a hexagonal crystal system and TeO₂ has a tetragonal crystal system. The feasibility of our new method has been proved in both of them. This method also works in very complicate system in theory, such as zeolite (Figure X1) and β -glumatic acid (supplementary Fig. 16). However, in practice, sometimes the e-beam damage or too small difference between two structures might bring difficulties to experiments. For example the chiral zeolites, we can also collect a tilt-series of STEM images while the resolution is limited (Figure X2). Without sub-angstrom resolution, the handedness cannot be determined by directly comparing two projections without markers.

Fig. 4 STEM-HAADF images of a tantalum silicide crystal along (a) [001] and (b) [-103] zone axes (insets are simulated (left) and *p*1 symmetry-averaged images (right) overlaid with structural models; purple spheres represent Ta atoms, Si atoms are omitted for clarify). (c) Projections of TaSi₂ crystal structural models with left-handedness along different orientations in a tilt series.

Figure X1. The structure projections of STW chiral zeolite along $[3-10]$ and $[310]$ are different. In principle, we can distinguish two enantiomorphic structures using STEM imaging by tilting a crystal from $[100]$ to $[3-10]/[310]$. The green rectangular marked an area for comparison.

Figure X2. ADF and ABF images taken from STW chiral zeolite along (a,b) $[100]$ and (c,d) $[1-10]$ directions. The insets are symmetry-averaged images overlaid with crystal structure models.

5. The determination of the handedness of the quartz crystal is not very convincing. In Figure 4, the simulated images are compared with the symmetry-averaged images. However, the agreement of the simulated and experimental images is rather poor. The aspect a/b ratio of the unit cell in the projection differs a lot between the experimental and simulated images for both Fig. 4a and 4b. The angle between a and b axes is ca 95 degrees in the experimental image in Fig. 4b, which deviated a lot from 90 degrees in the simulated images. Furthermore, the orientations of the black features corresponding to two SiO_4 tetrahedra are different in the simulated and symmetry-average images in Figure 4a. In Figure 4b, the contrast in symmetry-averaged image differ significantly from the experimental image. The authors should mention how the zone axes were determined, what symmetry has been applied and how this was done. Because the determination of the handedness is only

based on one image of poor quality here, and a slight error in atoms positions may change the bending direction, the conclusion is not very convincing.

Thanks for the reviewer's comment. The main reason for the poor quality of image is the e-beam instability of quartz under electron beam. We have test many crystals but the data in manuscript is still the best one. The determination of zone axes is based on indexing of SAED patterns (see supplementary Figure S10). We check the SAED patterns, the d-spacings of diffraction spots slightly deviate from the calculated ones using structural model, which might come from the inaccuracy of calibration. The angle deviation between a and b axes in the image might due to the image distortion during the scanning. The same angle calculated from the SAED pattern is very close to 90 degrees. The observation of 001 and 002 reflections is because the crystal is thick and we used the whole crystal for SAED. Moreover, the PXRD patterns were also collected to confirm the unit cell parameters and the phase of materials (supplementary Fig. 17), and the experimental data match well with the simulated one from quartz ($a = b = 4.921 \text{ \AA}$, and $c = 5.400 \text{ \AA}$). Therefore, there is no doubt that the crystal is quartz. The original image is low signal-to-noise ratio because of beam sensitivity. We have to use the very small spot size (spot size 10 in GrandARM300F) and the smallest condenser aperture (10 \mu m), around 0.9 pA . In this case, the signal is quite weak. To enhance the contrast, the symmetry-averaging was used because the space group of quartz is known, $P3_121$ or $P3_221$. The plane group symmetry $p2$ was applied to image along $[110]$ and pm to image along $[120]$. We totally understand the concerns of the reviewer. At present stage, unfortunately, we cannot get image with better resolution even using HRTEM imaging. Instead, we put the data from TaSi_2 as the new Figure 4 and moved quartz data to supplementary information.

Supplementary Fig. 10 Selected area electron diffraction (SAED) patterns of quartz taken along $[100]$ and $[210]$ zone axes from a single crystal in a tilt series.

6. Supplementary Figure 8 should include simulated image along $[110]$, which is relevant to the experimental image.

Thanks for the reviewer's comment. We have added the simulated image along [110] in supplementary Figure 13, which replaces the old supplementary Figure 8.

REVIEWERS' COMMENTS:

Reviewer #2 (Remarks to the Author):

The paper has been substantially modified, particularly through the addition of a new experimental example using TaSi₂ to replace the previous example using quartz which was less convincing. The quartz result has now been moved to supplementary information.

There are also additional simulations of other materials to which the method could potentially be applied in the supplementary information.

I think the modifications are appropriate and strengthen the paper. I am also happy with the changes made in response to my comments. I recommend publication.

Reviewer #3 (Remarks to the Author):

In the revised manuscript, the authors have addressed all my comments. The addition of the new example TaSi₂ makes me convinced about the method. I am now satisfied with the manuscript.

Therefore I recommend the publication of the manuscript in Nature Communications.